# Research on performance and dynamic competency evaluation of bid evaluation experts based on weight interval number

**Tie Li** [ORCID], **Guoliang Li***, **Mi Zhang, Yuan Qin, Guolong Wei**

Faculty of Civil Engineering and Mechanics, Kunming University of Science and Technology, Kunming, Yunnan Province, China

* liguoliang365@126.com

## Abstract

### Purpose/Significance

In the past many years, some scholars have studied bid evaluation experts, such as the behavior of bid evaluation experts. However, previous research ignores the performance and competency of bid evaluation experts, so this paper aims to provide a theoretical basis for incentive and constraint mechanism and hierarchical or dynamic management of bid evaluation experts by implementing performance and dynamic competency evaluation of bid evaluation experts.

### Method/Process

Firstly, the evaluation index system of performance and dynamic competency of bid evaluation experts is preliminarily constructed by referring to relevant literature, and then the constructed evaluation index was modified and improved by consulting relevant stakeholders' experts. Secondly, considering the hesitation and consistency of expert weighting, the calculation method of expert weight coefficient and index score interval number is improved. Based on the theory of weight interval number, the corresponding mathematical optimization model is constructed to calculate the index weight according to the purpose of performance judgment and dynamic competency clustering of bid evaluation experts. Finally, the data of performance and dynamic competency of bid evaluation experts is obtained by questionnaire survey, and the empirical analysis was carried out by simulating the bid evaluation experts consistent with the actual situation.

### Results/Conclusion

After improving the calculation method of index score interval number, and then calculating index weight interval number through index score interval number, the length of index weight interval number can be decreased and the calculation accuracy of index weight interval number can be increased. In addition, the index weight calculated by the constructed mathematical optimization model can make the intra-class discrimination smaller and the inter-

**Data Availability Statement:** All relevant data are within the paper.

**Funding:** Research on incentive mechanism of high-quality performance constraint of bid

evaluation experts Award Number: KKZ3202106005 Recipient:Guoliang LI.

**Competing interests:** The authors have declared that no competing interests exist.

class discrimination larger. Finally, some suggestions are also provided for the management of bid evaluation experts.

## 1. Introduction

Engineering bidding is a widely used transaction method in the world [1–3]. For project owners, the selection of contractors has a significant impact on project cost and quality [4]. In China, in order to select the bidders who best meets the bidding conditions, the relevant departments randomly select the bid evaluation experts in the relevant fields to form a temporary bid evaluation committee (China stipulates that the number of members of the bid evaluation committee is an odd number and more than 5 people) based on the provisions of relevant laws and regulations and the needs of the project, then the bid evaluation committee evaluates and selects the bidder according to the bidder's quotation, technical measures, etc. As a result, this leads to the inequality of legal responsibility and power of the bid evaluation subject, and the contradiction between the temporality of the bid evaluation committee and the long-term nature of the project. The two aspects become an important part of the research hotspot [1].

The evaluation and selection of contractors is a difficult and challenging task [5], and the decisions of bid evaluation and bid winning are often considered as key links in auctions [6]. Therefore, the assessment of contractors and the selection of best bidders require complex knowledge and experience to ensure that selected contractors are able to implement projects as required by owners [7]. At the same time, the bid evaluation committee can decide by itself, but not in an arbitrary way [8]. They hold the dominant power in the evaluation work and have the most direct and fundamental impact on the evaluation results [9]. Therefore, bid evaluation experts must have high competency. Although the relevant laws and regulations in China have stipulated the qualification of bid evaluation experts when they enter into expert database, there are no reliable measures to implement the periodic assessment system after the bid evaluation experts entered into expert database. Hence, the quality of bid evaluation experts is worrying [9]. Some provinces put forward the hierarchical or dynamic management of bid evaluation experts. In the long run, the ability of bid evaluation experts will change with the accumulation of knowledge and experience. Therefore, it is necessary to take periodic assessment the competency of bid evaluation experts as the theoretical basis of hierarchical management. In addition, the evaluation records of China's bid evaluation experts participating in the evaluation work are generally used for archiving and verification. Most provinces do not assess them through the performance of the bid evaluation experts. Although a few provinces propose "scoring system" management according to the performance of the bid evaluation experts, there are still large limitations, which only consider whether the bid evaluation experts are illegal or not.

At present, China is in the transition stage from offline bid evaluation (i.e. traditional bid evaluation method) to online bid evaluation. The bid evaluation process of the two methods is shown in Fig 1. Through comparison, the common points between the two and the advantages of online bid evaluation are found as follows:

(1) Common points between the two methods: no matter which method of bid evaluation is adopted, bid evaluation experts need to put forward bid evaluation suggestions, score the bids and put forward bid evaluation conclusions in the process of bid evaluation according to their professional knowledge and work experience. Meanwhile, they must comply with relevant laws and regulations.

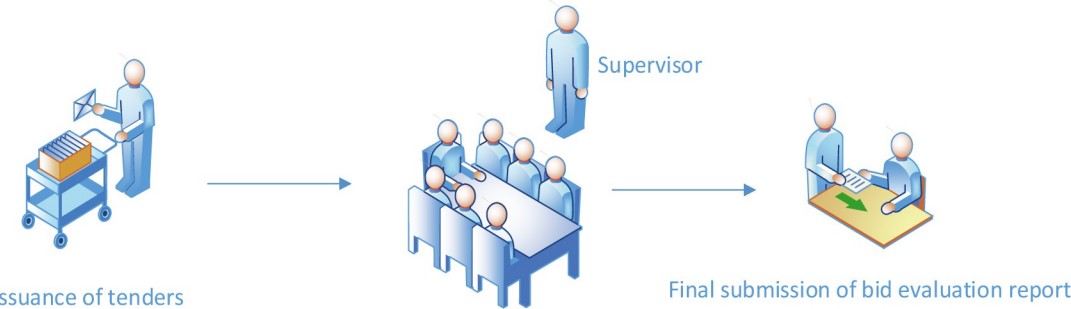

a. Offline bid evaluation

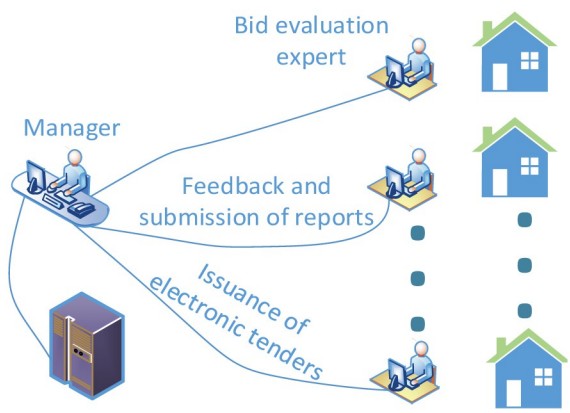

b. Online bid evaluation

**Fig 1. Two bid evaluation mechanisms.** a. Offline bid evaluation. b. Online bid evaluation.

(2) Advantages of online bid evaluation: no matter which method of bid evaluation is adopted, the performance of bid evaluation experts can be evaluated. However, online bid evaluation can automatically evaluate the performance of bid evaluation experts according to the evaluation process and results, and conduct periodic evaluation. At the same time, the digital footprint of the evaluation process of bid evaluation experts can be collected through technical means. Previous studies have shown that digital footprint provides an effective way to reduce information asymmetry and moral hazard [10–13]. Therefore, the performance of bid evaluation experts can be evaluated by digital footprint (such as the seriousness of performance of bid evaluation experts through equipment testing and the time for bid evaluation experts to browse bids, etc.). Online can realize off-site bid evaluation, experts do not need to meet, do not affect each other, and can realize incomplete information static game and independent bid evaluation.

Therefore, how to evaluate the performance of bid evaluation experts and periodically assess the competency of bid evaluation experts under the background of online bid evaluation

to provide a theoretical basis for the hierarchical or dynamic management and incentive and constraint mechanism of bid evaluation experts is a very meaningful research topic.

## 2. Related research review

### 2.1 Research on bid evaluation experts

The first section reviews the current management situation of bid evaluation experts in China and expounds the importance of competency and performance evaluation of bid evaluation experts and the limitations of current management under the background of information technology. The research on bid evaluation experts mainly includes two aspects: integrity, bid evaluation behavior and results. As for the integrity of bid evaluation experts, References [14, 15] constructed evaluation index system and evaluation model to evaluate the integrity of bid evaluation experts from different perspectives. As for the bid evaluation behavior and results of bid evaluation experts, the existing research focused on the behavior of the bid evaluation committee [16], the antagonism or uncooperative behavior of the bid evaluation expert groups (i.e. technical group and business group) [6, 17–20], the bid evaluation behavior [21] and collusive behavior [22] of bid evaluation experts as well as the abnormal score of bid evaluation experts [23] and the difference of score results [24]. In addition, References [25, 26] proposed incentivizing and constraining bid evaluation experts by analyzing the principal-agent relationship, and Reference [27] further designed the incentive and constraint mechanism. However, the consensus-building process of bid evaluation experts, the generation of collective decision-making matrix and the rank-oriented decision-making method consider the expert decision-making problem, which is different from the perspective of this paper and will not be further discussed in this paper.

The bid evaluation process of experts can be regarded as the expert service process. Due to the information asymmetry in the process of expert service, there are different kinds of hidden moral behavior in the expert service market, such as fraud, improper service, and internalization of the entire objective functions of the clients and so on [28]. Bid evaluation experts also have the characteristics of "gig economy" such as temporary feature and feature of project. Based on the network platform, the 'new gig economy' [29] in the Internet era has derived an online labor platform with algorithm as the underlying technical logic [30] by deeply integrating digital technology with the on-demand gig economy. Highly automated and data-driven method is used to replace the functions of managers in the labor execution management of platform workers through algorithm management [31], and the current situation of the incomplete and asymmetric information acquisition of both sides [32] and the principal-agent problem of incomplete labor contracts under the condition of asymmetric information are overcome through the exchange of a large amount of information [33]. It makes that the individual behavior of platform workers in the labor process is almost completely exposed to the continuous and rigorous monitoring environment of the algorithm. Therefore, they must show behavior consistent with organizational goals and platform specifications, and complete the assigned tasks [34, 35]. As a result, human resource management activities such as performance management are significantly different from traditional model [36]. By reshaping the work mode, the digital economy has triggered a series of new problems in behavior, efficiency and ethics in the workplace, so research on organizational behavior and human resource management at the micro level is urgently needed [37]. Big data and artificial intelligence simplify the data acquisition, and provide more research data that are difficult to obtain and trace for existing research [38]. They cover all aspects of the production process, and can penetrate into each production link, insight into relevant factors including human emotions and preferences

[39]. Therefore, it is feasible to evaluate the evaluation performance of online bid evaluation experts based on digital footprint research, which is also a topic worthy of in-depth discussion.

Reviewing the relevant research on bid evaluation experts, it is found that there is a lack of research on the performance evaluation of bid evaluation experts. In terms of China's relevant policy provisions, practical needs of management and theoretical research, it is urgent to study the performance evaluation of online bid evaluation experts, and consider that the changes in performance, integrity, knowledge, experience and other factors within a period may lead to changes in their competence, so as to provide support for management practice and expand relevant theories. According to the definition of performance, the performance of bid evaluation experts should comprehensively consider bid evaluation behavior and results. Referring to competency theory [40], competency should consider the performance and other related indicators within the period. Based on the dynamic view of competency theory [41] and the static and dynamic content characteristics of competency [42], this paper defines the periodic competency of bid evaluation experts as dynamic competency. In view of these, this paper will overall consider the relevant factors to evaluate the performance and dynamic competency of bid evaluation experts.

## 2.2 Research on subjective weighting method

Based on the above contents, this paper constructs the evaluation index system in order to realize the performance evaluation and dynamic competency evaluation of bid evaluation experts. Therefore, the calculation of reasonable and effective index weight has become the key issue of evaluation.

The common practice is to find some stakeholders (i.e. all those who have sufficient professional knowledge to carry out reasonable evaluation) [43] and combine the importance of the indices with the linguistic value to construct a judgment matrix through linguistic variables [44, 45]. In this way, the limitations of individual expert opinions can be avoided and the reliability of the evaluation results can be improved by integrating multiple expert opinions, such as analytic hierarchy process (AHP) [46] and order relationship analysis method(G1) [47, 48]. In this paper, IAHP method of reference [49] is used to construct the evaluation model. However, there are also some shortcomings in Reference [49]: Firstly, in the process of judging the importance of evaluation indices, the main professional knowledge characteristics (i.e. the hesitation of cognitive limitations and the consistency of different preferences) [50] as expert judgment information reflects the credibility of expert evaluation and affects the final evaluation results, while Reference [48] only considers the evaluation consistency to calculate the expert coefficient. Secondly, it is also unreasonable to eliminate expert coefficient in the calculation method of index score interval number constructed in Reference [49], because the size of expert coefficient represents the credibility of judgment results. Therefore, this paper improves the method proposed in Reference [49].

At the same time, the purpose of dynamic competency evaluation in this paper is to cluster bid evaluation experts and provide a theoretical basis for hierarchical management. Some scholars consider optimizing under the condition of calculating the weight interval number, including the highest satisfaction [51], the minimum weight deviation [52], and the minimum total projection deviation [53] as the optimization objectives. Therefore, this paper refers to this idea and constructs a mathematical optimization model under the condition of weight interval number.

The structure of this paper is as follows. Section 1 introduces the significance of performance and dynamic competency evaluation of bid evaluation experts. Section 2 reviews the relevant research on bid evaluation experts and the related theories of subjective weighting method. Section 3 constructs the evaluation index system of performance and dynamic competency of bid evaluation experts. Section 4 improves the calculation method of expert weight coefficient and

index score interval number based on the evaluation method proposed in Reference [49], and constructs a mathematical optimization model according to the purpose of performance and dynamic competency evaluation of bid evaluation experts. Section 5 makes an empirical analysis. Section 6 expounds the research conclusions of this paper, and puts forward suggestions for the management of bid evaluation experts based on the research and related theories.

## 3. Evaluation index system

### 3.1 Principles of constructing evaluation index system

1. Purpose principle. Realize the performance and dynamic competency evaluation of bid evaluation experts, and provide a theoretical basis for hierarchical management, incentive and constraint of bid evaluation experts.

2. Scientific principle. Fully follow the law of bid evaluation activities, and the selected indices, calculation methods and standards meet the characteristics of bid evaluation.

3. Practical principle. Conform to the objective reality, the selected index data are collectable and easy to operate.

4. Systematic principle. Comprehensively reflect the performance and dynamic competency of bid evaluation experts.

### 3.2 Construction process of evaluation index system

Based on the analysis of the management laws and regulations of some provincial bid evaluation experts and the current situation of bid evaluation of bid evaluation experts in China, this paper sorts the relevant evaluation indices of existing bid evaluation experts [14, 15], refers to other project evaluation experts [54, 55] and follow the above principles to preliminarily construct the evaluation index system of bid evaluation experts' performance and dynamic competency according to the common points of offline bid evaluation and online bid evaluation and the digital footprint of online bid evaluation. The performance evaluation index system includes 3 first-level indices, namely bid evaluation performance, bid evaluation quality and code of conduct, and 10 corresponding second-level evaluation indices. The dynamic competency evaluation index system includes 3 first-level indices of interim performance, code of conduct, and database-entry competency, and 8 corresponding evaluation indices. The expert consultation method is used to consult a total of 12 experts, which include 5 owners, 3 from regulatory agency, 2 from construction organization, and 2 bid evaluation experts, so as to modify and improve the evaluation indices, and finally construct the performance evaluation index system including 3 first-level indices of bid evaluation performance, bid evaluation quality, code of conduct, and the corresponding 11 second-level evaluation indices, as shown in Table 1. As well as the dynamic competency evaluation index system including the 2 first-level indices of the interim comprehensive situation, capacity improvement and the corresponding six second-level evaluation indices, as shown in Table 2. Finally, referring to the relevant references [54–56], the index calculation method is determined according to the actual situation of bid evaluation, as shown in Tables 1 and 2.

## 4. Evaluation model

### 4.1 Related theoretical knowledge

Definition 1 [58]. Let $X$ be a non-empty domain, then an intuitionistic fuzzy set $A$ on $X$ is: $A = \{\langle x, \mu_A(x), \nu_A(x)\rangle | x \in X\}$. In the formula, $\mu_A(x): X \rightarrow [0, 1]$ and $\nu_A(x): X \rightarrow [0, 1]$

**Table 1. Performance evaluation index system of bid evaluation experts.**

| First-level index | Second-level index | Second-level index remark | Calculation method |
|---|---|---|---|
| Bid evaluation performance $A_1$ | Study of bidding documents $B_1$ | Find unreasonable places in bidding documents and whether the suggestions are adopted | $C_1 = min(t_1, 9)$<br>t is the number of adopted suggestions, according to the number of suggestions made by experts and adopted, if suggestions do not match the browsing page, even if adopted will not score points |
| | Formal review $B_2$ | Find minor deviations in bid documents and confirm whether there are omissions after approval and post-qualification review | $C_2$ Refer to the above calculation method |
| | Responsiveness Review $B_3$ | Find significant deviation in bidding documents and whether they meets the relevant requirements in bidding documents | $C_3$ Refer to the above calculation method |
| | Detailed Review $B_4$ | Evaluate the bidding documents and put forward reasonable suggestions or find unreasonable parts in the bidding documents | $C_4$ Refer to the above calculation method |
| | Review Conclusion $B_5$ | Carefully fill in the review comments and review report, set bonus points for the situations that the constructive proposal of tenders is adopted and the together-conspired bidding is judged and recognized, and draw the evaluation conclusion (i.e., the order of the recommended bid winning candidates) | $C_5 = 0.5 \cdot min\{t_{51}, 9\} + 0.5 \cdot t_{52} \cdot 9$<br>$t_{52}$:SPEARMAN is rank correlation coefficient [41] |
| Quality of bidding evaluation $A_2$ | Score abnormality $B_6$ | Whether there is abnormal consistency in scoring (scoring by experts for different items of the same bid, scoring by experts for different bids, scoring among different experts), scoring errors or abnormally high or low, assignment of wrong scores, plagiarism. | $C_6 = max\{-t_6, -9\}$<br>$t_6$ is Number of unreasonable places |
| | Scoring credibility $B_7$ | Identify experts with significant bias effects on the evaluation data through the Tukey test, and experts are given additional points according to their credibility, the greater the credibility is, the more the additional points will be added | $C_7$ Calculating referring to Reference [45] |
| Code of conduct $A_3$ | Review Seriousness $B_8$ | Facial movement: Using facial information to analyze a person's concentration level through Facial Expression Recognition [57] | $C_8 = \{9,5,1\}$<br>9 Represents focused, 5 represents neutral, and 1 represents unfocused |
| | Timeliness $B_9$ | Time of submission of the review report and the review (i.e., the time to browsing each page of the bidding documents and the fitting of other experts) | $C_9 = max\left(9 - \frac{\sum_{i=1}^{m} \frac{\sigma_i}{t}}{m}, 0\right)$, $\bar{t} = \frac{1}{n}\sum_{i=1}^{n} t_i$, $\sigma_i^2 = \frac{1}{n}\sum_{i=1}^{n}(t_i - \bar{t})^2$<br>$\rho = 0$ or 1.0 indicates that not timely submission of review report. 1 means timely submission of bid evaluation report. $t_i$ represents expert evaluation time of the i-th page. $\bar{t}_i$ represents the average review time of i-th page of the same professional experts, $\sigma_i^2$ is the variance of expert review time, $n$ represents the number of professional experts, $m$ represents the number of pages of the bidding documents |
| | Discipline $B_{10}$ | Whether there are other circumstances stipulated by laws, regulations and rules such as not timely submission of review reports, imposture, disclosure of bid evaluation information, unauthorized departure from duty, use of communication tools, private contact with bidders, bribery, confirmation of participation in bid evaluation but not evaluating the bid without asking for leave. Action criteria: logging in to other web pages, using other applications, taking screenshots, photographing, etc. | $C_{10} = [0, -1, -3, -5, -7, -9]$<br>According to the impact on bid evaluation, if serious consequences are caused, bid evaluation qualification will be suspended (i.e. not providing bid evaluation information to the expert within a certain period of time) or cancelled. If there is an impostor, other indices do not score |

*(Continued)*

**Table 1.** (Continued)

| First-level index | Second-level index | Second-level index remark | Calculation method |
|---|---|---|---|
| | Strictness $B_{11}$ | Whether there are other situations stipulated by laws, regulations and rules, such as bid evaluation in strict accordance with the bid evaluation standards and methods of the bidding documents, calculation errors in bidding documents, etc. | $C_{11} = [0, -1, -3, -5, -7, -9]$<br>According to the influence on the bid evaluation, if serious consequences are caused, the suspension (i.e. not pushing the bid evaluation information to the expert within a certain period of time) or disqualification from bid evaluation |

are respectively the degree of affiliation and non-affiliation of element $x$ belonging to $A$, and satisfying $0 \le \mu_A(x) + \nu_A(x) \le 1$. $\pi_A(x) = 1 - \mu_A(x) - \nu_A(x)$ is the degree of hesitation of element $x$ in $A$, indicating degree of uncertainty that x belongs to the $A$, $0 \le \pi_A(x) \le 1$. All intuitionistic fuzzy sets on non-empty domain are denoted by $IFS(X)$, and $a = (\mu_a, \nu_a, \pi_a)$ is called intuitionistic fuzzy number ($IFN$), $\pi_a = 1 - \mu_A - \nu_A$ in this formula. The intuitionistic fuzzy number is expressed by $IFN$ in the following paper.

Definition 2 [59, 60]. Let $R$ denote a real number. If $a^-, a^+ \in R$ and $a^- \le a^+$, $a = [a^-, a^+]$ is called a binary interval number. If $a$ is a positive interval number, then $a = [a^-, a^+] = \{x \mid 0 \le a^- \le x \le a^+\}$.

## 4.2 Semantic information and intuitionistic fuzzy number

In this paper, referring to Reference [61], hesitation is divided into three levels of 'very small', 'small', 'general', where semantic evaluation granularity $r = 5$, $\pi = 0.1, 0.2, 0.3$ respectively represent three levels of hesitation, and the language evaluation value is quantified by referring to

**Table 2. Evaluation index system of dynamic competency.**

| First-level index | Second-level index | Second-level index remark | calculation method |
|---|---|---|---|
| Interim comprehensive situation $D_1$ | Interim performance $E_1$ | Set by the average value of performance in Table 1 | $G_1 = \frac{\sum_{h'_1} C_1^{h'_1}}{h_1}$<br>$C^{h'}$ is the performance of the $h'_1$-th bid evaluation and $h_1$ is the total number of bid evaluations |
| | Review Status $E_2$ | Complaint review after the end of the bid evaluation. | $G_2 = 9\left(1 - \frac{h'_2}{h_2}\right)$<br>$h'_2$ means the number of problems in review and errors in bid evaluation, $h_2$ means the number of review |
| | Participation $E_3$ | Set by participation rate | $G_3 = 9\frac{h'_3}{h_3}$<br>$h'_3$ is the number of participation in bid evaluation, $h_3$ is the number of receiving bid invitations. |
| | Assistance or cooperation with supervision, inspection $E_4$ | Assist or cooperate with the supervision and inspection of the relevant administrative supervision departments, set by good, comparatively good, average, comparatively poor and poor respectively. | $G_4 = \frac{\sum_{h'_4} C_4^{h'_4}}{h_4}$<br>$C^{h'_4}$ is the score of $h'_4$-th, taking 9, 7, 5, 3, 1. $h_4$ is the number of inspections |
| Competency improvement $D_2$ | Professional technical capability $E_5$ | Indicates the professional and technical ability of bid evaluation experts, including education background, scientific research ability, practical ability, etc. | Set $G_5$ by virtual professional technical ability |
| | Credit $E_6$ | Indicates the credit of bid evaluation experts, including personal credit, bid evaluation integrity, institution credit, etc. | Set $G_6$ by virtual credit |

**Table 3. Semantic information and IFNs.**

| Linguistic variables | Label | Intuitionistic fuzzy number (IFNs) | Quantitative value |
|---|---|---|---|
| Importance | I | $(0.8 - 0.5 \times \pi, 0.2 - 0.5 \times \pi)$ | 0.9 |
| Comparatively important | MI | $(0.7 - 0.5 \times \pi, 0.3 - 0.5 \times \pi)$ | 0.7 |
| Average | M | $(0.55 - 0.5 \times \pi, 0.45 - 0.5 \times \pi)$ | 0.5 |
| Comparatively unimportant | MUI | $(0.4 - 0.5 \times \pi, 0.6 - 0.5 \times \pi)$ | 0.3 |
| Unimportance | UI | $(0.3 - 0.5 \times \pi, 0.7 - 0.5 \times \pi)$ | 0.1 |

the reference [49, 62–64], as shown in Table 3. During the evaluation, $N$ experts independently evaluate the importance of the index $a_{M,i}$ of the $M(M \geq 2)$ layer on the upper level associated indices $a_{M-1,j}$.

## 4.3 Expert weight coefficient and index score interval number

The Reference [49] combined the basic theory of interval number with the hierarchical analysis method, proposed and proved the theorem of "positive interval number" and the theorem of "consistency of interval number judgment matrix". In the process of calculating the index score interval number, the expert weight coefficient is calculated considering the consistency of expert evaluation, and then the index score interval number is calculated according to the expert weight coefficient, but the expert weight coefficient is approximately subtracted in the calculation process. The actual calculated index score interval number is independent of the expert weight coefficient, and the calculation of expert weight coefficient only considers the consistency of expert evaluation, and does not consider the hesitation of expert evaluation, which is also incomplete. Therefore, this paper comprehensively considers the consistency and hesitation of expert evaluation to calculate the weight coefficient of expert evaluation, improves the method of calculating the index score interval number and proves that the improved calculation method meets the 'positive interval number' theorem. The improved calculation steps are as follows:

Step 1: Calculate the expert weight coefficient $\lambda_{n,i,j}^{M,1}$ [65] based on evaluation consistency according to the evaluation results of the importance of evaluation experts to indices.

$$\lambda_{n,i,j}^{M,1} = \frac{\frac{1}{1+\partial \varphi_{n,i,j}^{M}}}{\sum_{r=1}^{N}\left(\frac{1}{1+\partial \varphi_{n,i,j}^{M}}\right)} \tag{1}$$

In the formula, $\varphi_{n,i,j}^{M} = \left\{ \frac{1}{2}|\sigma_{n,i,j}^{M} - (\prod_{n=1}^{N}\sigma_{n,i,j}^{M})^{\frac{1}{N}}| + \varepsilon + \frac{1}{2}[\prod_{r=1,r\neq n}^{N}(|\sigma_{n,i,j}^{M} - \sigma_{r,i,j}^{M}| + \varepsilon)]^{\frac{1}{N-1}} \right\}$ [66] is the deviation coefficient. The larger the deviation is, the smaller the expert weight coefficient is, and the smaller the deviation is, the larger the deviation weight coefficient is. According to the Reference [66], the parameter $\partial$ is the adjustment coefficient. It is generally appropriate to define $\partial = 10$ in practical application. According to reference [65], $\varepsilon$ is a moderator variable with a value greater than 0. $\varepsilon = 0.2$ based on the standard characteristics of the index importance evaluation scale.

Step 2: Calculate the weight coefficient of experts based on hesitation $\lambda_{n,i,j}^{M,2}$ according to the evaluation results of the importance of evaluation experts to indices and IFNs. Due to the different professional knowledge and work experience of experts, there are different degrees of hesitation in the evaluation of the importance of the same index. In the judgment of the importance of a certain index, the greater the degree of hesitation is, the smaller the expert weight coefficient is, and the smaller the degree of hesitation is, the greater the expert weight

coefficient is.

$$\lambda_{n,i,j}^{M,2} = \frac{\frac{1}{\pi_{n,i,j}^{M}}}{\sum_{n=1}^{N} \frac{1}{\pi_{n,i,j}^{M}}} \tag{2}$$

Step 3: Calculate the expert weight coefficient $\lambda_{n,i,j}^{M}$ [61] in the evaluation based on comprehensively consideration of the consistency and hesitation of expert evaluation.

$$\lambda_{n,i,j}^{M} = \vartheta_1 \lambda_{n,i,j}^{M,1} + \vartheta_2 \lambda_{n,i,j}^{M,2} \tag{3}$$

In this formula, the parameters $\vartheta_1$, $\vartheta_2 \in [0,1]$ satisfy $\vartheta_1 + \vartheta_2 = 1$. When $\vartheta_1 > 0.5$, it indicates that more attention is paid to the consistency of expert evaluation information. When $\vartheta_2 > 0.5$, it indicates that more attention is paid to the determination of expert evaluation information. Since the evaluation experts are experts and scholars in this field, they are very familiar with each index. When evaluating, their hesitation is low and consistency information is more important, so $\vartheta_1 = 0.8$ and $\vartheta_2 = 0.2$ are determined.

Step 4: Calculate the index score interval number $P_{M,i}^{M-1,j}$.

$$P_{M,i}^{M-1,j} = [\Delta_{i,j}^{M} - \zeta_{i,j,1}^{M}, \Delta_{i,j}^{M} + \zeta_{i,j,2}^{M}] \tag{4}$$

The calculation method given in Reference [49] is as follows, and the reasons for improving it are also as follows.

$$\begin{cases} \Delta_{i,j}^{M} = \sum_{n=1}^{N} \lambda_{n,i,j}^{M} \sigma_{n,i,j}^{M} \\[2mm] \zeta_{i,j,1}^{M} = \frac{1}{\sum_n l_{n,i,j}} \sum_n [|\Delta_{i,j}^{M} - \sigma_{n,i,j}^{M}| \cdot l_{n,i,j}] \\[2mm] \zeta_{i,j,2}^{M} = \frac{1}{\sum_n (1 - l_{n,i,j})} \sum_n [|\Delta_{i,j}^{M} - \sigma_{n,i,j}^{M}| \cdot (1 - l_{n,i,j})] \\[2mm] l_{n,i,j} = \begin{cases} 1, \sigma_{n,i,j}^{M} \le \Delta_{i,j}^{M} \\ 0, \sigma_{n,i,j}^{M} > \Delta_{i,j}^{M} \end{cases} \end{cases}$$

Description:

$$\because \zeta_{i,j,1}^{M} = \frac{1}{\sum_n l_{n,i,j}} \sum_n [|\Delta_{i,j}^{M} - \sigma_{n,i,j}^{M}| \cdot l_{n,i,j}]$$

$$\therefore \text{ when } \sigma_{n,i,j}^{M} \le \Delta_{i,j}^{M}$$

$$\Delta_{i,j}^{M} - \zeta_{i,j,1}^{M} = \Delta_{i,j}^{M} - \frac{1}{\sum_n l_{n,i,j}} \sum_n [(\Delta_{i,j}^{M} - \sigma_{n,i,j}^{M}) \cdot l_{n,i,j}]$$

$$= \frac{1}{\sum_n l_{n,i,j}} \Big[ \Delta_{i,j}^{M} \cdot \sum_n l_{n,i,j} - \sum_n l_{n,i,j} (\Delta_{i,j}^{M} - \sigma_{n,i,j}^{M}) \Big]$$

$$= \frac{\sum_n l_{n,i,j} \cdot \sigma_{n,i,j}^{M}}{\sum_n l_{n,i,j}}$$

$$\therefore \Delta_{i,j}^{M} - \zeta_{i,j,1}^{M} = \frac{\sum_{n} l_{n,i,j} \cdot \sigma_{n,i,j}^{M}}{\sum_{n} l_{n,i,j}}, \text{ similarly, } \Delta_{i,j}^{M} + \zeta_{i,j,2}^{M} = \frac{\sum_{n}(1 - l_{n,i,j}) \cdot \sigma_{n,i,j}^{M}}{\sum_{n}(1 - l_{n,i,j})}$$

$\therefore$ This calculation method of interval number reduces the evaluation experts' weight coefficient $\lambda_{n,i,j}^{M}$ in the process of calculation, which is unreasonable.

In this paper, the improved calculation method is as follows. Firstly, it is proved that it satisfies the 'positive interval number' theorem, and then the rationality is explained.

$$\begin{cases} \Delta_{i,j}^{M} = \sum_{n=1}^{N} \lambda_{n,i,j}^{M} \sigma_{n,i,j}^{M} \\[2mm] \zeta_{i,j,1}^{M} = \sum_{n} \frac{\lambda_{n,i,j}^{M} \cdot l_{n,i,j}}{\sum_{n} \lambda_{n,i,j}^{M} \cdot l_{n,i,j}} \cdot |\Delta_{i,j}^{M} - \sigma_{n,i,j}^{M}| \\[2mm] \zeta_{i,j,2}^{M} = \sum_{n} \frac{\lambda_{n,i,j}^{M} \cdot (1 - l_{n,i,j})}{\sum_{n} \lambda_{n,i,j}^{M} \cdot (1 - l_{n,i,j})} \cdot |\Delta_{i,j}^{M} - \sigma_{n,i,j}^{M}| \\[2mm] l_{n,i,j} = \begin{cases} 1, \sigma_{n,i,j}^{M} \leq \Delta_{i,j}^{M} \\ 0, \sigma_{n,i,j}^{M} > \Delta_{i,j}^{M} \end{cases} \end{cases}$$

Proof: the interval number $P_{M,i}^{M-1,j}$ is a positive interval number.

$$\because l_{n,i,j} = 0 \text{ or } 1, \therefore 1 - l_{n,i,j} = 1 \text{ or } 0$$

$$\because \sigma_{n,i,j}^{M} > 0, \ \lambda_{n,i,j}^{M} > 0$$

$$\therefore \zeta_{i,j,1}^{M} \geq 0, \ \zeta_{i,j,2}^{M} \geq 0$$

$\therefore \Delta_{i,j}^{M} + \zeta_{i,j,2}^{M} - (\Delta_{i,j}^{M} - \zeta_{i,j,1}^{M}) = \zeta_{i,j,2}^{M} + \zeta_{i,j,1}^{M} \geq 0$, namely, $\Delta_{i,j}^{M} + \zeta_{i,j,2}^{M} \geq \Delta_{i,j}^{M} - \zeta_{i,j,1}^{M}$ (if and only if the scores of all evaluation experts are equal, the equal sign is reached and the interval number degenerates into a real number)

$$\because \Delta_{i,j}^{M} - \zeta_{i,j,1}^{M} = \Delta_{i,j}^{M} - \sum_{n} \frac{\lambda_{n,i,j}^{M} \cdot l_{n,i,j}}{\sum_{n} \lambda_{n,i,j}^{M} \cdot l_{n,i,j}} \cdot |\Delta_{i,j}^{M} - \sigma_{n,i,j}^{M}|$$

$$\therefore \text{ when } \sigma_{n,i,j}^{M} \leq \Delta_{i,j}^{M}$$

$$\begin{aligned} \Delta_{i,j}^{M} - \zeta_{i,j,1}^{M} &= \Delta_{i,j}^{M} - \sum_{n} \frac{\lambda_{n,i,j}^{M} \cdot l_{n,i,j}}{\sum_{n} \lambda_{n,i,j}^{M} \cdot l_{n,i,j}} \cdot (\Delta_{i,j}^{M} - \sigma_{n,i,j}^{M}) \\[2mm] &= \frac{1}{\sum_{n} \lambda_{n,i,j}^{M} \cdot l_{n,i,j}} \left[ \Delta_{i,j}^{M} \cdot \sum_{n} \lambda_{n,i,j}^{M} \cdot l_{n,i,j} - \sum_{n} \lambda_{n,i,j}^{M} \cdot l_{n,i,j} (\Delta_{i,j}^{M} - \sigma_{n,i,j}^{M}) \right] \\[2mm] &= \frac{\sum_{n} \lambda_{n,i,j}^{M} \cdot l_{n,i,j} \cdot \sigma_{n,i,j}^{M}}{\sum_{n} \lambda_{n,i,j}^{M} \cdot l_{n,i,j}} > 0 \end{aligned}$$

$\therefore \Delta_{i,j}^{M} + \zeta_{i,j,2}^{M} \geq \Delta_{i,j}^{M} - \zeta_{i,j,1}^{M} > 0$, the proof is completed.

Description: The rationality of the improved calculation method in this paper.

The calculation results of index score interval number given in Reference [49] are:

$$\Delta_{i,j}^M - \zeta_{i,j,1}^M = \frac{\sum_n l_{n,i,j} \cdot \sigma_{n,i,j}^M}{\sum_n l_{n,i,j}}, \quad \Delta_{i,j}^M + \zeta_{i,j,2}^M = \frac{\sum_n (1-l_{n,i,j}) \cdot \sigma_{n,i,j}^M}{\sum_n (1-l_{n,i,j})}$$

The calculation results of improved index score interval number in this paper are:

$$\Delta_{i,j}^M - \zeta_{i,j,1}^M = \frac{\sum_n \lambda_{n,i,j}^M \cdot l_{n,i,j} \cdot \sigma_{n,i,j}^M}{\sum_n \lambda_{n,i,j}^M \cdot l_{n,i,j}}, \quad \Delta_{i,j}^M + \zeta_{i,j,2}^M = \frac{\sum_n \lambda_{n,i,j}^M \cdot (1-l_{n,i,j}) \cdot \sigma_{n,i,j}^M}{\sum_n \lambda_{n,i,j}^M \cdot (1-l_{n,i,j})}$$

The expert weight coefficient reflects the credibility of the evaluation results. The greater the expert weight coefficient is, the higher the credibility is. It is found that the calculation method in Reference [49] reduces the expert weight coefficient $\lambda_{n,i,j}^M$ in the calculation process, which results in the calculation results are independent of the expert weight coefficient. The improved formula in this paper avoids this situation.

## 4.4 Evaluation model

According to the provisions of the bidding law, the number of members of the bid evaluation committee is an odd number and more than 5 people. In practice, the number of members of the bid evaluation committee is generally 5, 7, 9, which is not a large base. The purpose of performance evaluation is to judge the bid evaluation results of bid evaluation expert, so it is not necessary to distinguish performance of bid evaluation experts. The normalized weight vector of the index adopts the method of reference [49]. The purpose of periodic evaluation is to judge the change of competency of bid evaluation experts and realize the classification management of bid evaluation experts, which requires low discrimination of intra-class bid evaluation experts and a high discrimination of inter-class intra-class bid evaluation experts. Therefore, based on the calculation of the weight interval number, this paper determines the calculation method of the normalized weight vector of the index interval number according to the needs of performance evaluation and dynamic competency evaluation. The specific calculation process is as follows:

Step 1: Calculate the index score interval number $P_{M,i}^{M-1,j}$ by steps 1 to 4 in section 4.3, and then calculate the interval number judgment matrix [49].

$$P = (p_{ik})_{m \times m} \tag{5}$$

In the formula: $m$ is the index number of layer $M$ associated with index $a_{M-1,j}$ of layer $M-1$. $p_{ik}$ indicates the comparison result of the importance for $a_{M-1,j}$ between any two $a_{M,i}$, $a_{M,k}$ of layer $M$ associated with index $a_{M-1,j}$ of layer $M-1$, which is determined by formula (6).

$$p_{ik} = \frac{P_{M,i}^{M-1,j}}{P_{M,k}^{M-1,j}} = \frac{[\Delta_{i,j}^M - \zeta_{i,j,1}^M, \Delta_{i,j}^M + \zeta_{i,j,2}^M]}{[\Delta_{k,j}^M - \zeta_{k,j,1}^M, \Delta_{k,j}^M + \zeta_{k,j,2}^M]} = \left[ \frac{\Delta_{i,j}^M - \zeta_{i,j,1}^M}{\Delta_{k,j}^M + \zeta_{k,j,2}^M}, \frac{\Delta_{i,j}^M + \zeta_{i,j,2}^M}{\Delta_{k,j}^M - \zeta_{k,j,1}^M} \right] \tag{6}$$

Step 2: Transform the interval number judgment matrix into ordinary judgment matrices $P^L$ and $P^R$.

$$P^L = (p_{ik}^L)_{m \times m} \tag{7}$$

In the formula, $p_{ik}^L = \frac{\Delta_{i,j}^M - \zeta_{i,j,1}^M}{\Delta_{k,j}^M + \zeta_{k,j,2}^M}$, the matrix $P^L$ is the left matrix of the interval number judgment matrix $P$.

$$P^R = (p_{ik}^R)_{m \times m} \tag{8}$$

In the formula, $p_{ik}^R = \frac{\Delta_{i,j}^M + \zeta_{i,j,2}^M}{\Delta_{k,j}^M - \zeta_{k,j,1}^M}$, the matrix $P^R$ is the right matrix of the interval number judgment matrix $P$.

Step 3: Calculate the transfer matrices $A^L$, $A^R$ of $P^L$, $P^R$.

$$A^L = (lnp_{ik}^L)_{m \times m} \tag{9}$$

$$A^R = (lnp_{ik}^R)_{m \times m} \tag{10}$$

Step 4: Calculate the optimal transfer matrices $B^L$, $B^R$ of transfer matrices $A^L$, $A^R$.

$$B^L = (b_{ik}^L)_{m \times m} \tag{11}$$

$$B^R = (b_{ik}^R)_{m \times m} \tag{12}$$

In the formula, $b_{ik}^L = \frac{1}{m} \sum_{s=1}^{m} (lnp_{is}^L - lnp_{ks}^L)$, $b_{ik}^R = \frac{1}{m} \sum_{s=1}^{m} (lnp_{is}^R - lnp_{ks}^R)$.

Step 5: Calculate the quasi-optimal matrices $C^L$ and $C^R$ of $P^L$ and $P^R$ [67].

$$C^L = (c_{ik}^L)_{m \times m} \tag{13}$$

$$C^R = (c_{ik}^R)_{m \times m} \tag{14}$$

In the formula, $c_{ik}^L = 10^{b_{ik}^L}$, $c_{ik}^R = 10^{b_{ik}^R}$.

Step 6: Calculate the normalized vectors $W_j^L$ and $W_j^R$ of eigenvector corresponding to the largest eigenvalues $C^L$ and $C^R$, and obtain the weight interval number matrix $W_j^{LR}$ [68].

$$W_j^L = (w_{1,j}^{M,L}, \ w_{2,j}^{M,L}, \ldots, w_{m,j}^{M,L}) \tag{15}$$

$$W_j^R = (w_{1,j}^{M,R}, \ w_{2,j}^{M,R}, \ldots, w_{m,j}^{M,R}) \tag{16}$$

$$W_j^{LR} = ([\alpha w_{i,j}^{M,L}, \ \beta w_{i,j}^{M,R}])_{1 \times m} \tag{17}$$

In this formula, $\alpha$ and $\beta$ are determined by the following formulas.

$$\alpha = \sqrt{\sum_{k=1}^{m} \frac{1}{\sum_{i=1}^{m} c_{ik}^R}} \tag{18}$$

$$\beta = \sqrt{\sum_{k=1}^{m} \frac{1}{\sum_{i=1}^{m} c_{ik}^L}} \tag{19}$$

Step 7: Calculate normalized weight vector of performance evaluation index weight according to the formula in Reference [49], namely formulas (20), (21).

$$W_j' = \frac{\alpha W_j^L + \beta W_j^R}{2} \tag{20}$$

$$W_j = \frac{W_j'}{\sum_j W_j'} \tag{21}$$

The weight vector of dynamic competency evaluation index is calculated according to the goal of small intra-class discrimination and large inter-class discrimination. The smaller the

standard deviation is, the more concentrated the data is, indicating that the discrimination between the evaluation objects is smaller. Therefore, the intra-class discrimination is represented by standard deviation, and the inter-class discrimination is represented by deviation. The following mathematical optimization model is constructed to calculate the normalized weight vector of index weight.

Objective function:

$$\begin{cases} minV_1 = minSD^P_{z,z'} \\ maxV'_{z,z+1} = maxDE^P_{z,z+1} \end{cases} \tag{22}$$

Constraint conditions:

$$s.t. \begin{cases} \alpha w^{M,L}_{i,j} \leq w^{M'}_{i,j} \leq \beta w^{M,R}_{i,j} \\ w^M_{i,j} = \dfrac{w^{M'}_{i,j}}{\sum_{i=1}^m w^{M'}_{i,j}}, \text{ quasi } \sum_{i=1}^m w^M_{i,j} = 1 \\ G_p = \sum_M \prod_{j=1}^M w^M_{i,j} \cdot G^{M'}_{i,j,p} \end{cases}$$

Optimization method: The clustering is constantly updated to achieve optimization goal of minimizing the intra-class discrimination and maximizing the inter-class discrimination by iterating the weight value in the weight interval number.

In the formula, $G^{M'}_{i,j,p}$ represents the eigenvalue of the final layer index of the evaluation object $p$; $G_p$ and $G_q$ represent dynamic competency of the evaluation objects p and q; $DE^P_{z,z+1} = G_{p,z,min} - G_{p,z+1,max}$, and $G_{p,z,min} > G_{p,z+1,max}$. $V_z$ indicates the standard deviation of dynamic competency of the evaluation object within the $z$ class, $V'_{z,z+1}$ denotes the deviation of dynamic competency between $z$ and $z+1$ classes, then $W^M_j = (w^M_{1,j}, \ w^M_{2,j}, \ldots, w^M_{m,j})$ represents the normalized weight vector of the index interval number of the $M$ layer associated with the $M-1$ layer index $j$.

## 5. Empirical analysis

### 5.1 Calculation of index weight

**5.1.1 Performance of virtual bid evaluation experts.** In view of the particularity of the bid evaluation expert group, it is difficult to obtain relevant data. In order to make the performance and dynamic competency of virtual bid evaluation experts more realistic, this paper obtains some characteristics of the performance of bid evaluation experts through the expert survey of relevant departments, as shown in Table 4, and simulate the performance and dynamic competency of the virtual bid evaluation experts according to expert opinions.

(1) Performance of virtual bid evaluation experts

In this paper, 10,000 kinds of performance of bid evaluation experts are simulated as the basis for calculating the interim performance in the dynamic competency evaluation index of bid evaluation experts. In addition, 11 kinds of performance are randomly selected as the performance of 11 bid evaluation experts in a bid evaluation committee for one bid evaluation, which is used for empirical analysis, as shown in Table 5. The specific methods are as follows: firstly, analyze the dependency relationship among indices as noted in Table 4, determine an index with more dependency relationship among the indices with dependency relationship to generate, then generate other indices with dependency relationship, check the cross dependency relationship of indices, and correct the generated data with cross dependency

**Table 4. Characteristics of performance.**

| Performance | Index | General situation |
|---|---|---|
| Performance | Abnormality of ratings | 9 times and above 7.64%, 7 or 8 times 11.08%, 5 or 6 times 22.41%, 3 or 4 times 19.46%, 2 times and below 39.41% |
| | Score reliability | [0.9,1]36.32%, [0.8,0.9)28.93%, [0.7,0.8)21.72%, [0.6,0.7) 7.76%, [0,0.6)5.27% |
| | Seriousness of review | Focus state 47.43%, neutral state 42.27%, non-focus state 10.3% |
| | Sense of discipline | 2.88% |
| | Stringency | 2.99% |
| Dynamic competency | Situation of check | 90% and above 3.95%, [80%, 90%) 4.83%, [70%, 80%) 19.91%, [60%, 70%) 21.23%, below 60 50.08% |
| | Participation rate | 90% and above 41.47%, [80%, 90%) 19.80%, [70%, 80%) 15.88%, [60%, 70%) 12.75%, less than 60% 10.1% |
| | Assistance or cooperation in supervision, inspection | Very good 41.47%, good 18.96%, general 20.02%, poor 16.00%, very poor 3.55% |

Note: Individual indices not surveyed in evaluation are randomly assigned according to expert opinions. In dynamic competency, the interim performance is based on virtual 10000 kinds of performance, and the competency index is set according to the virtual value. The dependency relationship of performance evaluation indices is mainly that the better their code of conduct is, the better the bid evaluation performance and quality will be.

relationship, Finally, randomly combine the above indices with dependent relationship with the indices without dependent relationship.

The performance of bid evaluation experts is:

$$C = w_{10}(1 - \rho_1\rho_2)C_{10} + \rho_1\rho_2\sum_{i\neq10}w_iC_i \tag{23}$$

In the formula, $\rho_1 = 1$ or $0$, which indicate timely or not timely submission of bid evaluation report. $\rho_2 = 0$ or $1$, which indicates the existence or absence of impostor, $w_i$ indicates the index weight of final layer.

(2) Dynamic competency of virtual bid evaluation experts

The dynamic competency evaluation of bid evaluation experts is carried out on the basis of performance evaluation. In this paper, taking Kunming city as an example, there are about 1000 experts in the bid evaluation expert database in the field of engineering in Kunming according to survey. Setting up an evaluation cycle of 2 years, the number of experts drawn

**Table 5. Performance of 11 bid evaluation experts.**

| Index | $P_1$ | $P_2$ | $P_3$ | $P_4$ | $P_5$ | $P_6$ | $P_7$ | $P_8$ | $P_9$ | $P_{10}$ | $P_{11}$ |
|---|---|---|---|---|---|---|---|---|---|---|---|
| $B_1$ | 4 | 1 | 2 | 9 | 6 | 6 | 3 | 0 | 5 | 0 | 2 |
| $B_2$ | 6 | 6 | 5 | 2 | 0 | 8 | 3 | 5 | 1 | 6 | 2 |
| $B_3$ | 7 | 2 | 0 | 8 | 7 | 3 | 4 | 1 | 9 | 1 | 8 |
| $B_4$ | 2 | 7 | 6 | 4 | 8 | 4 | 6 | 9 | 3 | 7 | 7 |
| $B_5$ | 2.7 | 7.6 | 7.05 | 4.05 | 5.15 | 8 | 9 | 4.1 | 3.15 | 8.05 | 6.05 |
| $B_6$ | -2 | -4 | 0 | -5 | -1 | -3 | 0 | -1 | -6 | -2 | 0 |
| $B_7$ | 8.73 | 8.46 | 5.49 | 8.28 | 7.92 | 7.38 | 7.65 | 8.82 | 6.75 | 8.20 | 7.83 |
| $B_8$ | 9 | 5 | 5 | 1 | 9 | 5 | 9 | 9 | 5 | 1 | 5 |
| $B_9$ | 5.92 | 7.45 | 8.69 | 3.93 | 7.98 | 6.80 | 8.09 | 3.58 | 4.43 | 7.16 | 6.23 |
| $B_{10}$ | 0 | 0 | -1 | 0 | 0 | 0 | -5 | 0 | 0 | 0 | -3 |
| $B_{11}$ | 0 | -1 | 0 | 0 | -1 | 0 | 0 | 0 | -3 | 0 | 0 |

**Table 6. Dynamic competency of 10 bid evaluation experts.**

| Index | $P'_1$ | $P'_2$ | $P'_3$ | $P'_4$ | $P'_5$ | $P'_6$ | $P'_7$ | $P'_8$ | $P'_9$ | $P'_{10}$ |
|---|---|---|---|---|---|---|---|---|---|---|
| $E_1$ | 4.24 | 1.65 | 3.06 | 2.28 | 3.25 | 4.25 | 3.16 | 3.91 | 3.13 | 3.33 |
| $E_2$ | 7.52 | 5.73 | 7.79 | 8.61 | 3.60 | 7.20 | 5.40 | 5.40 | 3.60 | 2.70 |
| $E_3$ | 7.36 | 5.89 | 6.21 | 5.10 | 8.10 | 7.20 | 8.10 | 8.10 | 8.10 | 8.10 |
| $E_4$ | 7.00 | 7.00 | 9.00 | 7.00 | 3.00 | 1.00 | 5.00 | 1.00 | 5.00 | 9.00 |
| $E_5$ | 3.35 | 4.12 | 3.39 | 1.97 | 2.84 | 2.14 | 3.13 | 1.94 | 3.06 | 2.52 |
| $E_6$ | 2.70 | 0.72 | 2.21 | 1.57 | 1.40 | 1.05 | 0.01 | -1.18 | 1.06 | -0.19 |

accounts for 95% of the total number, the number of an expert drawn is about 1–100 times, and it is more likely to be drawn 10–20 times. Therefore, x (x∈[1,100]) times are extracted from 10,000 kinds of performance in line with the actual situation as the calculation basis of the interim performance in the dynamic competency, and x = 10–20 times are set as the times that most experts can be extracted in one cycle.

In addition, considering that performance evaluation is the basis of incentive and constraint mechanism of bid evaluation experts, it is assumed that the performance of bid evaluation experts in a cycle will not deteriorate under the effect of incentive and constraint mechanism, thus virtualizing the interim performance of bid evaluation experts in a cycle. The dynamic competency of a total of 1010 bid evaluation experts is virtualized, 1000 are used to calculate the index weight, and 10 were used for empirical analysis. Due to the limitation of space, only the dynamic competency of the 10 bid evaluation experts for empirical analysis is shown in Table 6.

**5.1.2 Index weight calculation.** Due to the different preferences of experts from relevant stakeholders on the performance evaluation indices, a total of 18 experts consisted of 4 owners, 3 from regulatory agency, 3 from construction organization, 3 from bidding agency, and 5 bid evaluation experts (3 experts from university and 2 experts from enterprise) judge the importance of the evaluation index (due to space limitations, some evaluation results are shown in Table 7).

The weight interval numbers of performance and dynamic competency evaluation indices calculated by formulas (1)–(19) are shown in Tables 7 and 8.

The normalized weight vector of performance evaluation index is calculated according to formulas (20) and (21), as shown in Table 9.

Through the optimization of formula (22), the calculated normalized weight vector of the final layer index of dynamic competency is shown in Table 10.

Through the calculation of the above weight interval number and index weight, it can be found that bid evaluation performance $A_1$ and bid evaluation quality $A_2$ have the same weight

**Table 7. Findings on the importance of expert segment indices.**

| Index<br>Expert | $A_1$ | $A_2$ | $A_3$ |
|---|---|---|---|
| 1 | I[1] | I[2] | I[1] |
| 2 | MI[1] | MI[1] | MI[2] |
| …… | …… | …… | …… |
| 17 | MI[3] | I[1] | I[1] |
| 18 | MI[1] | I[1] | M[2] |

Note: 1 Represents hesitation as 'very small ', 2 Represents hesitation as ' small ', 3 Represents hesitation as ' general '

**Table 8. Weight interval number of performance indices by improved method.**

| Matrix | Weight interval numbers |
|---|---|
| $(A_1, A_2, A_3)$ | ([0.3333,0.3478], [0.3333,0.3478], [0.3026,0.3333]) |
| $(B_1, B_2, B_3, B_4, B_5)$ | ([0.1777,0.2000], [0.2000,0.2056], [0.2000,0.2026], [0.2000, 0.2061], [0.2000,0.2080]) |
| $(B_6, B_7)$ | ([0.4904,0.5000], [0.5000,0.5096]) |
| $(B_8, B_9, B_{10}, B_{11})$ | ([0.2500,0.2748], [0.2384,0.2500], [0.2329,0.2500], [0.2500, 0.2539]) |

**Table 9. Weight interval number of dynamic competency indices by improved method.**

| Matrix | Weight interval numbers |
|---|---|
| $(D_1, D_2)$ | ([0.5000,0.5321], [0.4679,0.5000]) |
| $(E_1, E_2, E_3, E_4)$ | ([0.2707,0.3337], [0.2532,0.2707], [0.1358,0.1880], [0.2707, 0.2740]) |
| $(E_5, E_6)$ | ([0.4862,0.5000], [0.5000,0.5138]) |

**Table 10. Normalized weight vector of performance evaluation index.**

| Matrix | Weight |
|---|---|
| $(A_1, A_2, A_3)$ | (0.3410,0.3410,0.3180) |
| $(B_1, B_2, B_3, B_4, B_5)$ | (0.1889,0.2028,0.2012,0.2031,0.2040) |
| $(B_6, B_7)$ | (0.4952,0.5048) |
| $(B_8, B_9, B_{10}, B_{11})$ | (0.2624,0.2442,0.2414,0.2520) |

**Table 11. Normalized weight vector of final layer of dynamic competency.**

| Index | $E_1$ | $E_2$ | $E_3$ | $E_4$ | $E_5$ | $E_6$ |
|---|---|---|---|---|---|---|
| Weight | 0.1481 | 0.1353 | 0.0875 | 0.1399 | 0.2422 | 0.2470 |

interval number and the same index weights in the performance evaluation. The weight interval number of code of conduct $A_3$ is relatively close to the left side of bid evaluation performance $A_1$ and bid evaluation quality $A_2$, and its weight is also relatively close, indicating that experts from relevant stakeholders attach great importance to evaluation performance $A_1$, evaluation quality $A_2$ and code of conduct $A_3$, but pay more attention to evaluation performance $A_1$ and evaluation quality $A_2$.

In the dynamic competency evaluation, the interim comprehensive situation $D_1$ is on the right side of the competency improvement $D_2$, indicating that the experts of relevant stakeholders pay more attention to the interim comprehensive situation $D_1$ in the dynamic competency evaluation of bid evaluation experts, pay more attention to the interim performance $E_1$ in the interim comprehensive situation $D_1$, and pay more attention to the credit $E_6$ in the competency improvement $D_2$.

## 5.2 Comparative analysis

**5.2.1 Comparison of weight interval numbers.** Compared with the reference [49], the weight interval numbers of performance and dynamic competency evaluation indices calculated by the calculation method of Reference [49] are shown in Tables 11 and 12.

**Table 12. Weight interval number of performance evaluation indices calculated in reference [49].**

| Matrix | Weight interval numbers |
|---|---|
| $(A_1, A_2, A_3)$ | ([0.3333,0.3544], [0.3333,0.3544], [0.2913,0.3333]) |
| $(B_1, B_2, B_3, B_4, B_5)$ | ([0.1625,0.2000], [0.2000,0.2089], [0.2000,0.2049], [0.2000, 0.2105], [0.2000,0.2131]) |
| $(B_6, B_7)$ | ([0.4689,0.5000], [0.5000,0.5311]) |
| $(B_8, B_9, B_{10}, B_{11})$ | ([0.2500,0.2932], [0.2277,0.2500], [0.2170,0.2500], [0.2500, 0.2620]) |

**Table 13. Weight interval number of dynamic competency evaluation indices in reference [49].**

| Matrix | Weight interval numbers |
|---|---|
| $(D_1, D_2)$ | ([0.5000,0.5616], [0.4384,0.5000]) |
| $(E_1, E_2, E_3, E_4)$ | ([0.2700,0.3552], [0.2352,0.2700], [0.1333,0.1901], [0.2700, 0.2763]) |
| $(E_5, E_6)$ | ([0.4808,0.5000], [0.5000,0.5192]) |

**Table 14. Normalized ranking weight vector of final layer of dynamic competency.**

| Index | $E_1$ | $E_2$ | $E_3$ | $E_4$ | $E_5$ | $E_6$ |
|---|---|---|---|---|---|---|
| Weight | 0.1562 | 0.1354 | 0.0837 | 0.1408 | 0.2386 | 0.2453 |

**Table 15. Dynamic competency clustering in reference [49] and this paper.**

| Clustering | I | II | III | IV | V |
|---|---|---|---|---|---|
| Reference [49] | [4.0965,9] | [3.5771,4.0942] | [3.1622,3.5767] | [2.6825,3.1619] | [1.4515,2.6814] |
| Number of experts | 1322 | 2532 | 2845 | 2204 | 1097 |
| Length of interval | 4.9035 | 0.5171 | 0.4145 | 0.4794 | 1.2299 |
| Inter-class distance | — | 0.0003 | 0.0004 | 0.0003 | 0.0011 |
| This paper | [4.1457,9] | [3.5086,4.0148] | [3.1542,3.4584] | [2.7131,3.1529] | [1.5019,2.5329] |
| Number of experts | 1345 | 2486 | 2820 | 2234 | 1115 |
| Length of interval | 4.8543 | 0.5062 | 0.3042 | 0.4398 | 1.0310 |
| Inter-class distance | — | 0.1309 | 0.0502 | 0.0013 | 0.1802 |

Comparing the length *len* [69, 70] of each index weight interval number in Tables 7, 8, 11 and 12, it can be found that the length *len* of 22 interval numbers become small when the index weight interval number is calculated by the improved method. Therefore, the improved calculation method improves the calculation accuracy of the weight interval number and further proves the rationality of the improved calculation method of index score interval number.

**5.2.2 Comparison of clustering results.** After using the improved method to calculate score interval number of the dynamic competency evaluation index, then the index weight interval number is calculated (Table 8), and the normalized ranking weight vector of the final layer index then is calculated according to steps (16)—(22) of reference [49], as shown in Table 13.

Because the rating is generally set to 5 levels, the number of clusters is set to 5. The normalized ranking weight vector of the final layer index above (Table 13) and the optimization method are respectively used to cluster the dynamic competency of 10,000 virtual bid evaluation experts. The clustering interval of dynamic competency and the number of experts are

**Table 16. Comparison of clustering results of $P'_1 - P'_{10}$ bid evaluation experts' dynamic competency.**

| Experts | $P_1$ | $P_2$ | $P_3$ | $P_4$ | $P_5$ | $P_6$ | $P_7$ | $P_8$ | $P_9$ | $P_{10}$ |
|---|---|---|---|---|---|---|---|---|---|---|
| Dynamic competency calculated in Reference [49] | 4.7437 | 3.6717 | 4.6706 | 3.7896 | 3.1143 | 3.1511 | 3.3552 | 2.3359 | 3.3484 | 3.3860 |
| Dynamic competency calculated in this paper | 4.7469 | 3.6898 | 4.6764 | 3.7931 | 3.1282 | 3.1520 | 3.3665 | 2.3386 | 3.3617 | 3.3899 |
| Reference [49] Clustering | I | II | I | II | IV | IV | III | V | III | III |
| This paper clustering | I | II | I | II | IV | IV | III | V | III | III |

**Table 17. Comparison of the clustering discrimination of dynamic competency of 10 bid evaluation experts.**

| Discrimination | Intra-class discrimination | | | | Inter-class discrimination | | | |
|---|---|---|---|---|---|---|---|---|
| Comparison of Dynamic Competency discrimination with Normalized Ranking Weight Vector in Reference [49] | I | II | III | IV | I and II | II and III | III and IV | IV and V |
| | reduction 3.61% | reduction 12.40% | reduction 24.61% | reduction 35.48% | increase 0.26% | increase 4.96% | increase 6.30% | increase 1.45% |

obtained, and the length (*len*) of the clustering interval and the inter-class distance are calculated, the results are shown in Table 15 It can be found that the number of experts in each category is similar, and optimized clustering interval length (*len*) is smaller, and inter-class distance is larger. Therefore, the results are reliable when the weight interval number is optimized.

Through the above normalized ranking vector of the final layer index of dynamic competency (Table 13) and the normalized ranking weight vector of the final layer quality assurance of dynamic competency (Table 10), the dynamic competency of 10 bid evaluation experts ($P'_1 - P'_{10}$) is classified according to the clustering interval of this paper. The results are shown in Table 16.

According to the results of dynamic competency and clustering of 10 bid evaluation experts (Table 16), the reliability of optimization within the weight interval number is further proved.

**5.2.3 Comparison of clustering discrimination.** The goal of optimization is to minimize the intra-class discrimination and maximize the inter-class discrimination. According to the clustering results in Table 16, the intra-class discrimination and inter-class discrimination are compared by referring to formula (22), and the calculation results are shown in Table 17.

Through the data of Table 17, it can be found that the bid evaluation expert competency calculated in this paper has smaller intra-class discrimination and larger inter-class discrimination, which is conducive to the hierarchical management of bid evaluation experts in the expert database and the implementation of incentive and constraint mechanism. Therefore, the evaluation results of this paper are more in line with the actual needs.

## 6. Conclusions and suggestions

By constructing the evaluation index system and evaluation model of the performance and dynamic competency of bid evaluation experts, simulating bid evaluation experts accorded with the actual situation, and calculating the weight vectors of the performance and dynamic competency evaluation indices on the basis of the weight interval number, and finally carrying out the empirical analysis, the following conclusions and suggestions are drawn:

1. In the process of bid evaluation experts performing their duties, experts from relevant stakeholders attach great importance to the bid evaluation performance, bid evaluation quality and code of conduct of bid evaluation experts, but pay more attention to the bid evaluation performance and quality of bid evaluation experts.

2. In the dynamic competency evaluation of bid evaluation experts, experts from relevant stakeholders pay more attention to the interim comprehensive situation of dynamic competency evaluation of bid evaluation experts, pay more attention to the interim performance in the interim comprehensive situation, and pay more attention to credit in the competency improvement.

3. The improved calculation method of expert coefficient takes into account expert consistency and hesitation, which is more reasonable. The improved calculation method of index score interval number calculates the index score interval number and then calculates the weight interval number, which improves the calculation accuracy of the weight interval number, and the proposed mathematical optimization model meets the needs of hierarchical management of bid evaluation experts.

4. The proposed idea of optimization in weight interval numbers has good generality, which can also be used to set other optimization objectives or to evaluate other personnel.

5. The judgment results of the relevant stakeholders on the importance of evaluation indices reveal which aspects of quality of bid evaluation experts they pay more attention to, and also indicate which aspects of the bid evaluation experts may have prominent problems. Therefore, relevant management departments can strengthen the management in the future.

6. The bid evaluation experts participate in the project review after entering expert database, and carry out the 'scoring system' management through the performance evaluation (scoring according to the performance and the number of bid evaluation: high scores for good performance and low scores for poor performance. Each time they participate in the bid evaluation, scoring once, and accumulating the scores). After a cycle, the dynamic competency is re-evaluated and classified, and repeating the cycle, to achieve the purpose of hierarchical management and dynamic management of bid evaluation experts.

7. The relevant management departments may pay labor fees according to the performance of bid evaluation experts, give priority to the experts with high score and high competency to participate in project review, and kick experts with frequent poor performance out of the expert database.

This paper assumes that the performance of bid evaluation experts will not become worse under the effect of incentive and constraint mechanism is an ideal state. Referring to the performance curve of other staff under performance evaluation, the relationship between the performance of bid evaluation experts and the number of bid evaluations is complex. The performance curve may rise first and then tend to be stable, or it may be an inverted U-shaped curve. Future research can focus on the effect of incentive and constraint mechanism on the performance curve of bid evaluation experts to improve the reliability of virtual data.

## Acknowledgments

The authors thank Shuang Zheng, Yankun Peng, Tao Huang, Shaopeng Huang, Tonghai Li, Xianhai Qin, et al. for expert technical assistance.

## Author Contributions

**Data curation:** Tie Li.

**Funding acquisition:** Guoliang Li.

**Investigation:** Tie Li, Guolong Wei.

**Methodology:** Tie Li, Guoliang Li, Yuan Qin, Guolong Wei.

**Project administration:** Guoliang Li.

**Software:** Yuan Qin.

**Supervision:** Guoliang Li, Mi Zhang.

**Visualization:** Tie Li, Yuan Qin.

**Writing – original draft:** Tie Li, Mi Zhang.

**Writing – review & editing:** Tie Li, Mi Zhang.

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
