## [Decision Letter · Decision Letter 0]

22 Nov 2021

PONE-D-21-28448Research on performance evaluation of bid evaluation experts based on weight interval number theoryPLOS ONE

Dear Dr. Li,

Thank you for submitting your manuscript to PLOS ONE. After careful consideration, we feel that it has merit but does not fully meet PLOS ONE’s publication criteria as it currently stands. Therefore, we invite you to submit a revised version of the manuscript that addresses the points raised during the review process. Please take into account the reviewer's comments and suggestion to improve the paper.

We look forward to receiving your revised manuscript.

Kind regards,

Miguel Angel Sánchez Granero

Academic Editor

PLOS ONE

Journal Requirements:

5. We note you have included a table to which you do not refer in the text of your manuscript. Please ensure that you refer to Table 3 & 6 in your text; if accepted, production will need this reference to link the reader to the Table.

Reviewers' comments:

Reviewer's Responses to Questions

**Comments to the Author**

1. Is the manuscript technically sound, and do the data support the conclusions?

Reviewer #1: Partly

Reviewer #2: Yes

2. Has the statistical analysis been performed appropriately and rigorously? 

Reviewer #1: Yes

Reviewer #2: N/A

3. Have the authors made all data underlying the findings in their manuscript fully available?

Reviewer #1: Yes

Reviewer #2: Yes

4. Is the manuscript presented in an intelligible fashion and written in standard English?

Reviewer #1: No

Reviewer #2: Yes

5. Review Comments to the Author

Reviewer #1: This article presents a performance evaluation index system for bid evaluation experts and a corresponding evaluation model based on the calculation of the number of index weight intervals. However, I think the quality of the paper is not enough to meet the academic requirements, and there is still a lot of room for improvement. My comments to you generally represent a class of problems that the author needs to make serious revisions to the full text. Representative questions are as follows：

1. The novelty of the work is not very clear. The introduction does not reflect the importance of the research problem.

2. The literature review of the paper lacks systematicness and logic, the selection of references is not focused enough, and many papers are not related to the research topic of the paper.

3. The paper can not explain the superiority of the proposed method and can not reflect the innovation of the research work.

4. “Offline bidding evaluation and online bidding evaluation are two common bidding evaluation methods”, this statement is not correct!

5. Important statements lack literature support.

6. There are some grammar errors and statements are still not clarity. e.g. the last sentence in the first section of the introduction.

7. There are some format errors in Reference.

8. In the Description section, the writing format is not standardized.

9. The authenticity and reliability about data in Empirical Analysis need to clarified furtherly.

Reviewer #2: 1) The Abstract and Introduction sections should be improved. The information given by the manuscript is generally self-contained. However, there should be some improvements regarding its contents with necessary amplifications. There is lack of enough illustrations regarding the necessity of introducing performance evaluation of bid evaluation experts. I would suggest authors to provide a table to analyze the literature review. This will benefit the understanding the research gaps and your plans to fill them. The table will also help in comparing the existing models in the literature and highlighting its contributions in the literature. Please work on improving the clarity of your paper.

2) There are many existing publications in this research area. It is not clear the authors collected these papers based on which criteria. The review in the Introduction is too general. The historical review of the bid evaluation is weak. Authors are suggested to read some comprehensive and relevant publications on the existing counterparts to highlight the necessity of using weight interval number theory. For instance, some of them are proportional hesitant fuzzy linguistic term set and HFLTS possibility distribution, etc.

3) The level of English about this manuscript does not meet the journal's desired standard. Therefore, language should be greatly improved. There are too many grammatical mistakes and typos. Please carefully revise and improve it. The paper requires a thorough editing.

4) The author should state the source of the data and whether it is realistic data and how the expression of the presented measures for uncertainty being evaluated is useful for solving current real-life problems. More elaborations on these aspects are suggested.

5) The comparison analysis and in-depth discussions seem to be casual in this paper. Please enhance them to demonstrate the reliability of your advocated model. The related and recent work should be discussed and commented on. Some of them are: Bid evaluation in civil construction under uncertainty: A two-stage LSP-ELECTRE III-based approach and Bid Evaluation for Major Construction Projects Under Large-Scale Group Decision-Making Environment and Characterized Expertise Levels.

6) The managerial implications of this research should be enhanced. How decision or policymakers can benefit from this work with robust and reliable conclusions. What will change the main insights if different methods were introduced.

7) The conclusion should be improved to summarize clearly the main contributions of the paper and future research efforts. It will increase the impact of the paper if the authors try to indicate this explicitly in the manuscript. Critical limitations in the proposed framework should be offered. Extensions and applications of the proposal in other fields could be exemplified in the Conclusion section.

6. PLOS authors have the option to publish the peer review history of their article (what does this mean?). If published, this will include your full peer review and any attached files.

Reviewer #1: No

Reviewer #2: No

---

## [Author Response · Author response to Decision Letter 0]

13 Mar 2022

We responded to the journal editor and two reviewers in 'response to reviewers' ，thanks！

---

## [Decision Letter · Decision Letter 1]

4 Apr 2022

PONE-D-21-28448R1Research on performance and dynamic competency evaluation of bid evaluation experts based on weight interval numberPLOS ONE

Dear Dr. Li,

Thank you for submitting your manuscript to PLOS ONE. After careful consideration, we feel that it has merit but does not fully meet PLOS ONE’s publication criteria as it currently stands. Therefore, we invite you to submit a revised version of the manuscript that addresses the points raised during the review process.

I suggest the author to take into account the reviewer's comments.

We look forward to receiving your revised manuscript.

Kind regards,

Miguel Angel Sánchez Granero

Academic Editor

PLOS ONE

Journal Requirements:

Reviewers' comments:

Reviewer's Responses to Questions

**Comments to the Author**

1. If the authors have adequately addressed your comments raised in a previous round of review and you feel that this manuscript is now acceptable for publication, you may indicate that here to bypass the “Comments to the Author” section, enter your conflict of interest statement in the “Confidential to Editor” section, and submit your "Accept" recommendation.

Reviewer #1: All comments have been addressed

Reviewer #2: All comments have been addressed

2. Is the manuscript technically sound, and do the data support the conclusions?

Reviewer #1: Yes

Reviewer #2: Yes

3. Has the statistical analysis been performed appropriately and rigorously? 

Reviewer #1: Yes

Reviewer #2: N/A

4. Have the authors made all data underlying the findings in their manuscript fully available?

Reviewer #1: Yes

Reviewer #2: Yes

5. Is the manuscript presented in an intelligible fashion and written in standard English?

Reviewer #1: Yes

Reviewer #2: Yes

6. Review Comments to the Author

Reviewer #1: Compared with the previous version, the paper has been improved greatly, but there is still much room for improvement in some areas. The literature review of the paper is not sufficiently relevant and focused compared with the core issues of the paper. Management inspiration and conclusion recommendations are put together.

Reviewer #2: The authors respond well to my comments, I think it is greatly improved and is ready for publication in the current form.

7. PLOS authors have the option to publish the peer review history of their article (what does this mean?). If published, this will include your full peer review and any attached files.

Reviewer #1: No

Reviewer #2: No

---

## [Author Response · Author response to Decision Letter 1]

18 May 2022

Dear editors and reviewers, we carefully considered your suggestions and made revisions. We uploaded the 'Response to Reviewers' for detailed responses.

---

## [Decision Letter · Decision Letter 2]

23 May 2022

Research on performance and dynamic competency evaluation of bid evaluation experts based on weight interval number

PONE-D-21-28448R2

Dear Dr. Li,

We’re pleased to inform you that your manuscript has been judged scientifically suitable for publication and will be formally accepted for publication once it meets all outstanding technical requirements.

Kind regards,

Miguel Angel Sánchez Granero

Academic Editor

PLOS ONE

Additional Editor Comments (optional):

Reviewers' comments:

Reviewer's Responses to Questions

**Comments to the Author**

1. If the authors have adequately addressed your comments raised in a previous round of review and you feel that this manuscript is now acceptable for publication, you may indicate that here to bypass the “Comments to the Author” section, enter your conflict of interest statement in the “Confidential to Editor” section, and submit your "Accept" recommendation.

Reviewer #1: (No Response)

2. Is the manuscript technically sound, and do the data support the conclusions?

Reviewer #1: Yes

3. Has the statistical analysis been performed appropriately and rigorously? 

Reviewer #1: Yes

4. Have the authors made all data underlying the findings in their manuscript fully available?

Reviewer #1: Yes

5. Is the manuscript presented in an intelligible fashion and written in standard English?

Reviewer #1: Yes

6. Review Comments to the Author

Reviewer #1: (No Response)

7. PLOS authors have the option to publish the peer review history of their article (what does this mean?). If published, this will include your full peer review and any attached files.

Reviewer #1: No

---

## [Editor Report · Acceptance letter]

13 Jun 2022

PONE-D-21-28448R2 

Research on performance and dynamic competency evaluation of bid evaluation experts based on weight interval number 

Dear Dr. Li:

I'm pleased to inform you that your manuscript has been deemed suitable for publication in PLOS ONE. Congratulations! Your manuscript is now with our production department. 

Kind regards, 

on behalf of

Dr. Miguel Angel Sánchez Granero 

Academic Editor

PLOS ONE